# The Intersection of Sarcopenia and Musculoskeletal Pain: Addressing Interconnected Challenges in Aging Care

**DOI:** 10.3390/ijerph22040547

**Published:** 2025-04-02

**Authors:** Yacov Grosman, Leonid Kalichman

**Affiliations:** 1Department of Physical Therapy, Recanati School for Community Health Professions, Faculty of Health Sciences, Ben-Gurion University of the Negev, P.O. Box 653, Beer Sheva 84105, Israel; yacovgrosman@gmail.com; 2Department of Physical Therapy, Meuhedet Health Maintenance Organization, Rosh Haayin 4809139, Israel

**Keywords:** sarcopenia, musculoskeletal pain, aging population, integrated healthcare, multidisciplinary care

## Abstract

The global aging population faces a growing prevalence of sarcopenia and musculoskeletal (MSK) pain, two interrelated conditions that diminish physical function, quality of life, and independence in older adults. Sarcopenia, characterized by the loss of muscle strength, mass, and function, often coexists with MSK pain, with emerging evidence suggesting that each condition may contribute to the progression of the other. This perspective explores the bidirectional relationship between sarcopenia and MSK pain, highlighting shared mechanisms, including inactivity, cellular aging, chronic inflammation, gender-related hormonal changes, and psychosocial factors such as depression and social isolation, which underlie the mutual exacerbation between conditions. Through a multidisciplinary framework, the article emphasizes integrating care across specialties to address these interconnected conditions. Practical approaches, including comprehensive screening protocols, tailored resistance exercise, and nutritional support, are discussed alongside innovative hybrid care models combining in-person and telemedicine systems to enhance accessibility and continuity of care. A call to action is presented for clinicians, policymakers, and researchers to adopt collaborative strategies, prioritize investment in integrated healthcare, and bridge critical knowledge gaps. By reframing care delivery and advancing multidisciplinary efforts, this perspective aims to effectively address the complex challenges posed by the intersection of sarcopenia and MSK pain in older adults.

## 1. Introduction

People worldwide are living longer. The percentage of the global population ages 65 and above is projected to rise from 10% in 2022 to 16%, reaching 1.5 billion by 2050, and the number of adults aged 80 or older is expected to triple between 2020 and 2050 to reach 426 million [1]. Longer lifespans achieved through medical advancements do not always translate to healthy life years. Often, they are accompanied by disability, an increased risk for chronic diseases, and a diminished quality of life [2]. These challenges produce a heavy economic and psychosocial burden for patients and significantly strain healthcare system budgets [3]. Among the chronic conditions that significantly affect older adults, sarcopenia and musculoskeletal (MSK) disorders stand out due to their substantial impact on daily functioning and overall well-being [4].

Sarcopenia, characterized by a gradual loss of muscle mass, strength, and function, often leads to adverse health outcomes, including increased susceptibility to falls, fractures, and a consequent decline in independence [5]. Concurrently, MSK disorders, encompassing a wide range of conditions affecting the muscles, bones, and joints, are predominant sources of chronic pain and disability among older adults [6]. The interaction between sarcopenia and MSK pain can worsen the severity of both conditions, creating a cycle of declining physical function, mental health challenges, and social isolation, potentially accelerating the loss of independence among the elderly [7].

Despite being distinct conditions, sarcopenia and MSK pain frequently coexist in older adults, where their shared mechanisms and bidirectional effects amplify the burden of both conditions. However, clinical care and research often fail to address the interplay between these entities, focusing instead on isolated symptoms rather than the interconnected biological, psychological, and social factors that drive their interaction. This fragmented understanding limits our ability to develop effective, integrated strategies for prevention and management.

This perspective offers a novel approach to addressing these gaps by adopting a multidisciplinary framework to explore the shared mechanisms linking sarcopenia and MSK pain. We propose a conceptual model to elucidate their bidirectional relationship and identify opportunities for intervention. By advancing this understanding, we aim to establish a foundation for innovative care approaches that better meet the complex needs of older adults.

## 2. Approach and Scope of the Perspective

*Literature Search Strategy:* This perspective article presents a narrative synthesis based on an extensive literature search conducted across electronic databases, including PubMed, Scopus, EBSCO, and Google Scholar, to identify articles relevant to sarcopenia and MSK disorders in aging populations. The search incorporated a combination of keywords and Medical Subject Headings (MeSH) terms, such as “sarcopenia”, “musculoskeletal disorders”, “chronic pain”, “aging”, “muscle mass”, and “muscle strength”, employing Boolean operators (e.g., AND, OR) and wildcard characters to refine the scope. The search phrases included the following: “sarcopenia AND musculoskeletal disorders”, “sarcopenia AND chronic pain”, “aging AND sarcopenia”, “aging AND musculoskeletal disorders”, and “aging AND chronic pain”. The search process was iterative, with terms adjusted based on initial findings to ensure the inclusion of pertinent studies. The search was restricted to articles published in English from inception to July 2024 to encompass historical developments and recent advancements in the field.

*Selection Criteria:* Given the narrative nature of this perspective, articles were selected based on their relevance to epidemiology, pathophysiology, clinical implications, and management strategies concerning sarcopenia, MSK disorders, and their interconnectedness in aging populations. Studies were critically evaluated, prioritizing the hierarchy of evidence in evidence-based medicine [8]. Randomized controlled trials, cohort studies, systematic reviews, and meta-analyses were particularly emphasized to provide robust support for the discussed concepts. The relevant literature identified through this targeted search was critically evaluated and narratively integrated to clearly articulate the biological mechanisms, psychosocial factors, clinical implications, and innovative interventions central to the intersection of sarcopenia and musculoskeletal pain. Data from selected studies were analyzed thematically, emphasizing key concepts and emerging themes to support the integrative insights and recommendations presented within this perspective article.

## 3. Skeletal Muscle: Significance and Age-Related Changes

Understanding skeletal muscle’s systemic functions is critical to appreciating how its decline with age contributes not only to sarcopenia but also to musculoskeletal pain and broader health deterioration. Skeletal muscles are vital for daily functional performance and overall health, as in a healthy adult, it account for about 40% of total body mass [9]. Beyond locomotion and thermoregulation, it functions as a key reservoir of amino acids and carbohydrates, supporting organ-specific protein synthesis in tissues such as the skin, brain, and heart, and contributes substantially to basal energy expenditure [10,11]. Given the essential functions of skeletal muscle in various physiological processes, any deterioration in its quality and mass can have profound implications.

The aging process is linked with a decline in muscle mass and strength, which can commence as early as the fourth decade of life [12]. Adults lose approximately 25% of their individual peak muscle mass between the ages of 40 and 70, with muscle loss accelerating after age 70 with a gradual decline of about 2% each year [9,12,13]. Muscle strength declining with aging (dynapenia) is much more rapid and occurs at a 2–5 times faster rate than the concomitant loss of muscle mass, suggesting a decline in muscle quality [14,15,16]. Specifically, individuals aged 75 and above typically experience a decline of around 60% in their peak muscle strength and approximately 30% in their physical function [12]. At the intersection of the age-related decline in skeletal muscle mass and strength lies a specific condition termed ‘sarcopenia’. While the natural aging process can lead to some loss of muscle function, sarcopenia represents a more severe and clinically significant deterioration. These age-related changes in skeletal muscle are foundational to understanding its role in the interplay between sarcopenia and MSK pain.

## 4. Sarcopenia

Irwin Rosenberg coined the term sarcopenia in the late 1980s, deriving it from the Greek words “sarx”, meaning flesh, and “penia”, signifying loss. Initially introduced to characterize the age-related decline in muscle mass, sarcopenia is now recognized as a distinct disease condition, reflecting decades of research and understanding [17]. A significant milestone in this evolution was the awarding of the International Classification of Diseases, Tenth Revision, Clinical Modification (ICD-10-CM) code (M62.84) to sarcopenia in 2016 [18]. This formal recognition of sarcopenia as a disease entity has been crucial for fostering further research, clinical diagnosis, and treatment, thereby contributing to the growing body of knowledge and interventions to manage this condition.

Sarcopenia is defined as a progressive skeletal muscle condition marked by an accelerated decline in strength, muscle mass, and physical function with increasing age [5,18]. While the condition is well recognized, there is ongoing discussion about the optimal definition and standardized clinical evaluation of sarcopenia. The most commonly referenced definition is the revised version presented by the European Working Group on Sarcopenia in Older People (EWGSOP2) in January 2019 [19]. According to this definition, sarcopenia is diagnosed in individuals exhibiting low muscle strength and either low muscle mass or quality. In practice, sarcopenia is suspected when low muscle strength is identified, with the diagnosis confirmed by the presence of low muscle quantity or quality. In cases where low muscle strength, low muscle quantity/quality, and diminished physical performance are all evident, sarcopenia is classified as severe [18,19].

Besides the widely recognized EWGSOP2 criteria, several other frameworks have been proposed to diagnose sarcopenia. Notable among these are the definitions provided by the Foundation for the National Institutes of Health (FNIH), the Asian Working Group for Sarcopenia (AWGS), the Sarcopenia Definitions and Outcomes Consortium (SDOC), and the International Working Group on Sarcopenia (IWGS) [20,21,22,23]. Although these frameworks differ in specific criteria, they generally focus on three key domains: muscle strength, typically assessed via hand grip strength; muscle mass, often measured using appendicular skeletal muscle mass adjusted to height or body mass index; and physical performance, commonly evaluated through gait speed [19,20,21,22,23]. This variability reflects the challenge of addressing sarcopenia across different populations and clinical settings. Advancing these efforts is crucial not only for accurate diagnosis and management but also for understanding sarcopenia’s broader role in the interplay with MSK pain and functional decline.

## 5. Epidemiology and Health Implications of Sarcopenia

The prevalence of sarcopenia varies considerably across different populations, regions, and age groups. This variation is primarily attributed to the different diagnostic criteria used across studies, leading to prevalence rates ranging from less than 10% to as high as 40% of adults [24]. Geographic and ethnic characteristics further influence these rates, reflecting the muscle mass and strength variability across different ethnic groups [24,25]. Furthermore, the method used to evaluate muscle mass and strength significantly influences the estimated prevalence of sarcopenia. Variances in muscle mass thresholds and assessment techniques, such as Bioelectrical Impedance Analysis versus Dual-Energy X-ray Absorptiometry (DXA), may explain some differences in prevalence estimates observed across studies [26]. Additionally, it is suggested that the prevalence of severe sarcopenia may be influenced by the physical performance measures used, such as gait speed tests. Specifically, longer walk course lengths for a given gait speed cut-point have been associated with a decrease in estimated prevalence [24]. Furthermore, comorbidities significantly influence the prevalence of sarcopenia. A strong association has been found between probable sarcopenia and several long-term conditions, including MSK, endocrine, neurological, psychiatric, eye, and cardiovascular conditions [27]. Particularly, confirmed sarcopenia has been linked to a diverse group of pathologies, including type 2 diabetes mellitus [28], respiratory disease [29], cardiometabolic disease [30], and increased risk of falls and fractures [31,32]. These associations highlight the systemic nature of sarcopenia and its implications beyond muscle function alone. Furthermore, sarcopenia contributes to significant adverse outcomes, including loss of physical independence [33], cognitive impairment and depression [34,35], lower quality of life [36], and all-cause mortality [37].

These findings emphasize the need for standardized diagnostic criteria to enable more accurate estimates of sarcopenia prevalence across diverse populations. Moreover, recognizing the complex interplay between sarcopenia, comorbidities, and influencing factors is crucial for advancing research into its shared mechanisms with MSK pain and developing more effective management strategies.

## 6. Pathophysiology and Risk Factors of Sarcopenia

The pathophysiological process underlying the age-related decline in muscle mass and strength is complex and multifactorial. While certain mechanisms and pathways of skeletal muscle loss with aging remain elusive and not fully understood, specific factors contributing to this decline have been identified.

Aging disrupts skeletal muscle homeostasis, which hinges on a delicate balance between protein synthesis and breakdown [5]. This disruption is accentuated by shifts in anabolic hormones, including testosterone, growth hormone, and insulin-like growth factor-1. As levels of these hormones decrease with age, muscle protein synthesis becomes impaired. This leads to an imbalance between muscle protein anabolic and catabolic pathways, resulting in a net loss of skeletal muscle [38]. Beyond hormonal changes, aging also impacts muscle tissue’s metabolic and cellular environment. Alterations in insulin signaling [39] and mitochondrial function [40] contribute to this decline. Chronic low-grade inflammation and oxidative stress damage muscle cells, hampering their repair and regeneration processes [41]. Additionally, the age-related decline in muscle satellite cells, which are essential for muscle repair and regeneration, further diminishes the muscles’ ability to regenerate [42]. Consequently, this reduces muscle growth and recovery capacity, compromising muscle function over time [38].

Concurrently, neuromuscular degeneration also plays a pivotal role in sarcopenia. The progressive loss of motoneurons leads to a reduction in muscle fiber number and size. The decline in muscle function arises from the insufficient reinnervation of muscle fibers by the remaining motoneurons [43]. Moreover, age-related structural and functional changes at the neuromuscular junction, the synapse between the motor neuron and muscle fiber, can hinder neural signal transmission to muscles, resulting in decreased muscle activation and force production [39]. As a result of these combined factors, sarcopenic muscle is characterized by a reduction in the size and number of myofibers, especially affecting type II fibers. This leads to a decreased number of motor units and an accumulation of fat within muscle tissue [39]. The reduced muscle activation and force production, in turn, contribute to the overall decline in muscle strength and functionality observed in sarcopenia.

Beyond neuromuscular degeneration, chronic inflammation, and hormonal shifts, genetic predisposition and lifestyle habits are also recognized as significant risk factors for the onset and progression of sarcopenia [42]. Suboptimal nutrition, particularly insufficient protein intake, can hinder muscle protein synthesis, thereby facilitating muscle wasting [44]. Furthermore, a decline in physical activity and a predominantly sedentary lifestyle are directly associated with muscle mass deterioration and functional decline. Physical inactivity and muscle disuse lead to muscle deterioration and amplify the age-related effects previously discussed, intensifying the overall decline in muscle health [39]. In fact, the compounded effects of disuse and aging can be particularly detrimental to muscle structure and function. Strong evidence suggests that disuse may be the prime cause of the aging-related loss of muscle mass and strength [45,46]. Exploring these interconnected mechanisms and risk factors offers valuable insights into sarcopenia’s broader role in MSK pain. It supports the development of interventions addressing shared biological and lifestyle contributors to functional decline.

## 7. MSK Disorders in the Aging Population

In addition to the increased rate of sarcopenia, chronic MSK disorders also rise with age, presenting a significant health concern for the elderly population [47]. According to the proposed ICD-11 classification, chronic MSK pain is described as persistent or recurrent pain lasting longer than three months, originating from disease processes directly affecting bones, joints, muscles, or related soft tissues [48]. This definition predominantly falls under the nociceptive category, arising from actual or potential tissue damage in the MSK system. However, not all pain perceived in these tissues directly originates from them. Some manifestations, such as compression neuropathy or somatic referred pain, have different causes but manifest in the MSK regions [49]. Recognizing this distinction is imperative for accurate diagnosis and targeted interventions.

In the context of older adults, it is essential to consider the broad spectrum of MSK pain experiences. Given the age-related complexities and the potential for multiple coexisting health conditions in this population, a comprehensive understanding is vital. This encompasses disorders characterized by persistent inflammation, such as rheumatoid arthritis; those involving structural changes, like symptomatic osteoarthritis; and conditions with less clear origins, such as nonspecific back pain [50]. Older adults are particularly susceptible to chronic MSK pain due to age-related biological changes such as altered pain processing, increased comorbidities, reduced physical resilience, and cumulative mechanical stress over time [47,51]. Understanding the diverse nature of these conditions is crucial for targeted interventions and underscores the global burden these disorders represent. Their impact is far-reaching, affecting millions of individuals worldwide and placing a significant strain on healthcare systems.

Data from the Global Burden of Disease 2019 highlight that approximately 1.71 billion people globally live with MSK conditions, which are the leading contributor to disability worldwide [52]. The prevalence of common MSK conditions significantly correlates with age. In the elderly population, persistent MSK pain rates range from 40% to 60% [6]. Furthermore, most studies focus on MSK pain at a single site. However, localized pain is not the reality for many patients, and pain in more than one anatomical site is common, especially among older adults. Studies show that about 25–43% of community-dwelling adults aged 65 and older report MSK pain in two or more sites, representing a significant burden in this growing population [53,54].

MSK disorders in the elderly population have profound implications for overall health, well-being, and quality of life. The occurrence of MSK disorders among older adults is associated with frailty, depression, cognitive impairment, falls, poor sleep quality, and increased social isolation [50,55,56,57]. Many of these associated problems may arise from and lead to reduced physical activity, which is a prevalent risk factor associated with various health conditions such as diabetes, cardiovascular diseases, and cancer as leading causes of mortality [58].

Multisite pain within this population further amplifies these challenges. It exacerbates the effects of MSK disorders by further hindering mobility, increasing fall risk, and intensifying psychological issues such as depression and anxiety, which contribute to social withdrawal [59,60,61]. Moreover, the cumulative burden of multisite pain is correlated with functional decline and disability, highlighting the urgent need for comprehensive management strategies [62]. Addressing the rising burden of MSK disorders in aging populations requires a deeper exploration of their shared mechanisms with sarcopenia. This approach could inform more integrative and effective interventions, particularly for those facing multisite pain and its cascading effects.

## 8. Relationship Between MSK Pain and Sarcopenia

### 8.1. The Bidirectional Cycle

MSK pain and sarcopenia are prevalent conditions among the elderly, with aging being a significant factor in their onset [7]. While each has been extensively studied, the complex and potentially bidirectional relationship between them is an emerging area of interest. Recent studies have begun to shed light on the specific connections between the conditions suggesting higher rates of sarcopenia in patients with MSK pain, particularly with spinal disorders and low back pain [63,64], shoulder pain [65], knee osteoarthritis, and knee pain [66]. These findings underscore the need to explore how these conditions interact and perpetuate one another.

Persistent MSK pain can substantially limit mobility and engagement in regular physical activity, promote a sedentary lifestyle, and increase vulnerability to malnutrition, depression, and social isolation [7,67,68,69]. Each of these pain-related complications is recognized as contributing to sarcopenia’s progression, with physical inactivity being one of the most direct causes [44,46,70]. This can create a vicious cycle where prolonged pain leads to decreased physical activity, resulting in reduced muscle mass, which in turn exacerbates pain and further diminishes activity levels. For instance, chronic low back pain is a recognized risk factor that contributes to decreased physical activity and disability [71], often accompanied by inadequate nutrient intake [72], both of which are key contributors to the onset and progression of sarcopenia.

Emerging evidence also suggests that sarcopenia can amplify MSK pain [73]. It inherently involves diminished muscle mass and strength, heightening the risk of injuries and amplifying MSK pain, particularly in joints that bear weight and the spinal region [74,75]. Preliminary findings suggest that reduced lean mass may be linked to increased pain sensitivity, particularly in mechanically stressed areas like the lower back and shoulders. However, more extensive studies are needed to confirm these associations [76].

Postural deviations, such as anterior pelvic tilt and increased thoracic kyphosis, are commonly observed in individuals with sarcopenia. These misalignments place additional mechanical strain on the MSK system, potentially contributing to chronic pain in regions like the lower back, neck, and shoulders [77]. Sarcopenia also affects postural stability, increasing the risk of falls and MSK injuries [78]. The deterioration of postural control mechanisms contributes to increased strain on both weight-bearing and non-weight-bearing joints, which may amplify MSK pain [79].

Beyond its effects on general MSK alignment and stability, emerging evidence highlights the growing recognition of sarcopenia’s role in pelvic floor dysfunction. The decline of pelvic floor muscle function due to sarcopenia is linked to conditions such as pelvic organ prolapse, incontinence, and other dysfunctions [80], which are often linked to pelvic MSK pain [81]. The altered biomechanics resulting from pelvic muscle weakness can also lead to compensatory movements and postural adjustments, potentially affecting other areas of the body, such as the lower back and hips [82].

This intricate relationship underscores the mutual influence of MSK pain and sarcopenia, where the exacerbation of one condition reciprocally aggravates the other, creating a cycle of worsening pain, muscle deterioration, and functional decline.

Delving deeper into this complex cyclical relationship between chronic MSK pain and sarcopenia, it becomes evident that both conditions contribute to decreased physical function and activity in older adults [83,84]. This is further supported by the observation that older adults with persistent MSK pain often adopt a sedentary lifestyle, leading to muscle disuse and accelerating age-related sarcopenia [85]. The resulting decline in physical activity not only exacerbates muscle wasting but also precipitates other health conditions, such as cardiovascular disease and metabolic syndrome [86], which can further compound the effects of sarcopenia and MSK pain, thereby creating a feedback loop where each condition potentiates the other.

Thus, MSK pain leads to inactivity, which directly accelerates sarcopenia while also contributing to systemic health complications that indirectly exacerbate muscle loss. Conversely, sarcopenia is associated with broader health issues, including cardiometabolic disorders [30], which can indirectly worsen MSK pain through intermediary mechanisms such as inflammation, insulin resistance, and increased fat mass [87,88].

These interconnected pathways underscore the need for integrated approaches to manage these interrelated conditions effectively. Thus, addressing the bidirectional cycle between MSK pain and sarcopenia requires a multidisciplinary approach that targets both shared mechanisms and the broader health complications arising from their interplay.

### 8.2. Sociodemographic and Gender-Related Factors

Apart from these cyclical interactions, sarcopenia and MSK pain also share several sociodemographic risk factors. Building upon the previously mentioned determinants, both conditions appear to be influenced by factors such as female sex, low socioeconomic status (SES), and low educational level, highlighting the broader social and environmental contexts in which these conditions occur [5,7,89].

Gender-related factors are particularly relevant when examining the shared risk between sarcopenia and MSK pain. Women, particularly postmenopausal women, are at higher risk for both conditions, with some research suggesting that hormonal changes, such as the decline in estrogen, could contribute to reductions in muscle quality and MSK pain [90,91]. Estrogen is known to influence muscle protein synthesis and degradation, both critical for maintaining muscle health and preventing muscle deterioration that can lead to MSK pain [92]. Studies indicate that estrogen may inhibit muscle atrophy during periods of disuse and stimulate muscle regeneration, which is important for preserving muscle mass and preventing the MSK strain that can exacerbate pain [93]. Additionally, lower estrogen levels are linked to increased proinflammatory cytokines, which promote muscle breakdown and inflammation-related pain, worsening both sarcopenia and MSK pain [94,95]. However, it is important to note that there is limited research on the influence of estrogen on muscle in vivo, and this relationship is complex, likely also influenced by lifestyle factors, socioeconomic conditions, and healthcare access [85]. Moreover, women often report higher levels of chronic pain, which may be connected to biological, psychosocial, and environmental factors [96]. Understanding the interplay between hormonal changes and these broader gender-related factors is crucial for developing targeted interventions for sarcopenia and MSK pain.

In addition to gender-related factors, low SES and low educational level are key sociodemographic risk factors influencing both sarcopenia and MSK pain. Individuals with lower SES may experience limited access to healthcare, reduced health literacy, and fewer opportunities for regular physical activity, all of which contribute to the onset and progression of these conditions [97,98,99]. Poor nutrition, particularly insufficient intake of protein and essential micronutrients, exacerbates muscle wasting and weakens bones, further linking low SES to sarcopenia and MSK pain [100].

A low educational level further compounds these effects, as reduced health literacy limits awareness of preventive measures such as balanced nutrition, physical activity, and early medical intervention [97,98,101]. Income disparities may also restrict access to quality healthcare, rehabilitation services, and adequate nutrition [102,103]. These financial constraints, coupled with occupations that are either physically demanding or sedentary, increase the risk of muscle deterioration and chronic pain [98,101]. Addressing these sociodemographic factors is critical to reducing disparities in the management and prevention of sarcopenia and MSK pain.

### 8.3. Biological and Physiological Mechanisms

To better understand the intersection of MSK pain and sarcopenia, it is essential to examine the underlying biological and physiological mechanisms that may contribute to their development and progression. Among these, low-grade chronic inflammation emerges as a critical shared pathway, influencing both conditions. Chronic low-grade inflammation is characterized by a persistent, systemic inflammatory response that is typically mild but can significantly contribute to the pathophysiology of both sarcopenia and MSK disorders [7]. Elevated levels of inflammatory cytokines, such as tumor necrosis factor-alpha (TNF-α), interleukin-6 (IL-6), and C-reactive protein (CRP), have been consistently observed in older adults with sarcopenia [104]. These cytokines promote protein degradation pathways, inhibiting muscle protein synthesis and impairing muscle regeneration, leading to increased muscle protein breakdown and reduced synthesis, exacerbating muscle atrophy and weakness [41,104]. Similarly, chronic MSK pain conditions, such as osteoarthritis, rheumatoid arthritis, and fibromyalgia, are associated with elevated levels of inflammatory cytokines. These cytokines contribute to pain sensitization, cartilage degradation, and joint inflammation [105,106]. Such inflammatory processes affect both the peripheral and central nervous systems, leading to chronic pain syndromes and heightened pain sensitivity. In the peripheral nervous system, inflammatory cytokines like TNF-α, IL-6, and IL-1β sensitize nociceptors, which are pain receptors in peripheral tissues. This sensitization triggers the release of additional inflammatory mediators, increasing nociceptor sensitivity and excitability of pain pathways [107]. Concurrently, these inflammatory processes promote synovial inflammation and joint destruction, perpetuating pain in MSK conditions like osteoarthritis [108]. Prolonged exposure to inflammatory mediators can also lead to central sensitization, where neurons in the spinal cord and brain become hypersensitive to pain stimuli. This results in amplified pain signals and a reduced pain threshold, characterized by persistent pain even after the initial injury or inflammation has resolved, contributing to chronic pain syndromes and functional limitations [105].

Chronic inflammation also affects muscle regeneration by inhibiting muscle satellite cells, which are crucial for muscle repair and growth. The presence of chronic inflammations disrupts satellite cell activation and proliferation, leading to reduced muscle repair and regeneration capabilities [109]. This impairment not only contributes to the progression of sarcopenia but may also exacerbate MSK pain by altering the distribution and function of satellite cells, hindering the recovery of injured muscle tissues, and further contributing to pain and functional limitations [110,111,112].

Additionally, chronic inflammation increases oxidative stress, further damaging satellite cells and diminishing their regenerative capacity, exacerbating both muscle atrophy and MSK pain [113,114]. Age-related mitochondrial dysfunction, which is characterized by compromised energy supply and elevated intracellular oxidative stress, plays a significant role in these processes by impairing cellular metabolism and increasing the production of reactive oxygen species, potentially contributing to the decline in muscle function and exacerbation of pain [114]. This combination of oxidative stress and mitochondrial dysfunction accelerates muscle loss and amplifies inflammatory processes, creating a vicious cycle that links sarcopenia and MSK pain [115]. As inflammation, oxidative stress, and mitochondrial dysfunction interact, they mutually reinforce the decline in muscle health and the persistence of chronic pain. These interconnected mechanisms highlight the need to address shared biological pathways when developing targeted interventions for sarcopenia and MSK pain. Recognizing and mitigating these mechanisms could provide new opportunities for breaking the muscle wasting and pain cycle in older adults.

### 8.4. Clinical Variability

While the biological and physiological processes provide substantial evidence for the interplay between sarcopenia and MSK pain, clinical findings reveal a more nuanced and inconsistent relationship. It is crucial to highlight that, to date, the research on the clinical relationship between these conditions is limited in scope, and the findings from these studies often present conflicting results. For instance, a cross-sectional study of 730 Japanese community-dwelling older adults reported that low back pain was strongly associated with frailty status but not with sarcopenia [116]. A three-year prospective study reported that women with prevalent chronic knee pain experienced greater declines in lower-limb strength than the pain-free participants, but no chronic pain effects were observed on sarcopenia-related outcomes in men [117]. Conversely, in a prospective Health, Aging, and Body Composition study, appendicular lean mass and grip strength were associated with the risk of clinically diagnosed symptomatic knee OA and knee pain among men but not women [66].

These inconsistencies are further illustrated in a recent meta-analysis, where a positive correlation was observed between sarcopenia and MSK pain. However, this association was significantly influenced by the location of the pain. While patients with low back pain exhibited a high prevalence of sarcopenia, no positive association was found between lower-limb pain and sarcopenia [7]. This finding is intriguing, especially considering that lower-limb pain is a major risk factor for physical inactivity. Thus, these discrepancies point to the complex and multifaceted nature of the sarcopenia–MSK pain relationship, which is shaped by diverse biological, psychological, and social factors.

In summary, the intricate relationship between sarcopenia and MSK pain is influenced by a variety of biological, physiological, and lifestyle factors (Figure 1). Chronic inflammation, mitochondrial dysfunction, and oxidative stress play pivotal roles in linking these conditions, while clinical findings highlight the variability and complexity of their interplay. Recognizing the bidirectional nature of this relationship is essential for developing effective therapeutic strategies.

## 9. Implications for Patient Care

The complex interplay between sarcopenia and MSK pain underscores the need for innovative, multidisciplinary strategies that target shared biological, psychological, and social pathways. Effective care requires collaboration among diverse professionals, with each addressing specific aspects of the bio-psycho-social framework within their expertise.

### 9.1. Comprehensive Screening Approach

In contrast to MSK pain, sarcopenia remains critically underdiagnosed across healthcare settings despite its profound impact on health outcomes [118]. A comprehensive and collaborative approach to screening is essential, involving all healthcare providers to improve early detection and intervention. For individuals presenting with MSK pain, screening for sarcopenia offers an opportunity to identify underlying muscle weakness or mass deficits that may contribute to pain severity and functional decline. Similarly, identifying MSK pain in individuals with sarcopenia may highlight barriers to maintaining physical activity or adequate nutrition, providing critical insights for tailored interventions. This integrated approach allows healthcare providers to address the shared risk factors and interdependent mechanisms of both conditions, improving outcomes through early, targeted management.

Primary care physicians often serve as the first point of contact and are instrumental in identifying patients at risk for sarcopenia. Through clinical suspicion guided by observable signs such as unintentional weight loss, reduced physical activity, or impaired strength and knowledge of contributory conditions like metabolic imbalances or prolonged immobilization, physicians can initiate referrals for further evaluation by geriatric specialists or physical therapists (PTs). Given the well-documented interplay between sarcopenia and MSK pain, PTs are uniquely positioned to integrate sarcopenia screening into routine MSK assessments. As a cornerstone of MSK pain treatment [6,119], PTs regularly perform functional evaluations such as gait analysis, strength testing, and mobility assessments, aligning with the diagnostic criteria for sarcopenia.

Other healthcare professionals, including dietitians, psychologists, and social workers, contribute valuable perspectives to the screening process. Dietitians address nutritional deficits that are both risk factors for sarcopenia and potential consequences of chronic MSK pain, such as inadequate protein intake or micronutrient deficiencies. Psychologists and social workers play critical roles in identifying psychosocial factors, including chronic stress, depression, or social isolation, that contribute to the progression of both sarcopenia and MSK pain.

Collaborative screening protocols that leverage the expertise of multiple disciplines are essential to bridging the gap between sarcopenia and MSK pain. By aligning efforts across healthcare providers, early diagnosis becomes a powerful tool in preventing the escalation of these interconnected conditions. Such an integrated approach lays the foundation for timely, targeted interventions that address shared mechanisms and improve patient outcomes.

### 9.2. Addressing Cyclical Contributing Factors

Healthcare providers should recognize and address the cyclical nature of sarcopenia and MSK disorders, where one condition can exacerbate the other. Breaking this feedback loop requires targeted strategies that address shared risk factors and the interdependent mechanisms of these conditions. Given the significant overlap in the underlying mechanisms of sarcopenia and MSK disorders, interventions aimed at one condition could potentially benefit the other:Promote Physical Exercise: Early and progressive engagement in tailored resistance exercise programs has shown promise in reversing muscle loss associated with sarcopenia and alleviating chronic MSK pain [120,121]. PTs play a critical role here by developing individualized exercise plans that are pain-sensitive and functionally oriented, enabling patients to regain mobility and strength without exacerbating symptoms;Nutritional Support: Nutritional interventions should be integral to the management plan. MSK pain can lead to reduced appetite and inadequate nutrient intake, which accelerates muscle wasting [72]. Conversely, sarcopenia-related physical limitations can restrict access to nutrient-rich foods, particularly in older adults with mobility challenges [100]. Nutritional interventions tailored to optimize protein intake, address micronutrient deficiencies, and support energy balance are critical for mitigating these interactions;Target Chronic Inflammation: Low-grade chronic inflammation serves as a shared biological pathway linking sarcopenia and MSK disorders. Elevated levels of proinflammatory cytokines contribute to muscle protein degradation and pain sensitization, exacerbating both conditions. Interventions such as resistance exercise, anti-inflammatory dietary approaches, and, where appropriate, pharmacological treatments can help attenuate systemic inflammation, breaking the cycle of muscle wasting and chronic pain [122,123,124];Address Psychosocial Factors: Psychosocial barriers, often overlooked, require equal attention. Chronic pain and sarcopenia can lead to depression, social isolation, and fear of movement, further reducing physical activity and worsening the cycle [125,126]. Moreover, both sarcopenia and chronic MSK pain have been independently associated with cognitive decline in older adults, likely due to shared pathways such as inflammation, reduced physical activity, malnutrition, and psychosocial distress [127,128]. Physicians, psychologists, and social workers each play an important role in recognizing and addressing cognitive and psychosocial barriers through early identification, behavioral support, and interventions that help patients manage challenges and maintain function;Strengthen Medical Integration Across Care: Given the strong association of sarcopenia and MSK pain with systemic conditions such as metabolic syndrome, diabetes, and cardiovascular disease, medical doctors (MDs) play a crucial role in comprehensive care. MDs can evaluate and manage systemic contributors to these conditions, such as insulin resistance, systemic inflammation, and medication side effects, ensuring that the broader health picture is addressed. By coordinating with PTs, dietitians, and mental health professionals, MDs help integrate medical management with functional and psychosocial interventions. This collaboration ensures that targeted therapies, whether pharmacological or lifestyle-based, align with patients’ overall health needs, optimizing outcomes for sarcopenia and MSK pain management.

To effectively address the cyclical and interconnected nature of sarcopenia and MSK pain, innovative approaches that transcend traditional care models are essential. While targeted interventions such as resistance exercise, nutritional support, and psychosocial care play a pivotal role, their implementation is often limited by logistical and systemic barriers. Bridging these gaps requires embracing advancements in healthcare delivery, particularly through telemedicine, to enable integrated multidisciplinary care that is both accessible and adaptable to individual patient needs.

### 9.3. Leveraging Telemedicine for Multidisciplinary Care

While traditional in-person care remains the gold standard, it can be limited by factors such as clinic accessibility, transportation difficulties, and the availability of rehabilitation personnel. These barriers are especially pronounced for community-dwelling older adults with multiple chronic conditions, who may face logistical, economic, and mobility challenges and who may struggle to attend regular appointments [129]. To bridge the gap between clinical recommendations and real-world implementation, telemedicine offers a transformative solution for delivering integrated, multidisciplinary care to address sarcopenia and MSK pain. By leveraging digital platforms, telemedicine enables collaboration among healthcare providers [130], improves access to care, and promotes patient adherence to comprehensive treatment plans, particularly for underserved populations such as older adults with mobility challenges or limited access to rehabilitation services [131].

Tele-exercise serves as a cornerstone of this approach. Physical therapists can remotely deliver personalized resistance exercise programs, ensuring patients receive targeted, progressive interventions to improve muscle mass, strength, and pain management [132]. By leveraging digital platforms and wearable devices, tele-exercise enhances adherence and allows real-time monitoring and adjustments. These tools help maintain continuity of care and address workforce shortages while ensuring patients remain engaged [133].

Beyond physical therapy, telemedicine fosters interdisciplinary collaboration by enabling dietitians, psychologists, and social workers to provide remote consultations and interventions. For example, dietitians can guide patients in optimizing nutritional intake to support muscle health and manage pain-related fatigue [134,135]. At the same time, psychologists can address the fear of movement and pain-related anxiety through virtual therapy sessions [136]. This digital integration ensures that the bio-psycho-social dimensions of sarcopenia and MSK pain are comprehensively addressed.

Hybrid care models, combining in-person and telemedicine approaches, exemplify the potential of this approach. Initial in-person assessments can establish a clinical baseline, while follow-up care is delivered virtually to maximize convenience without compromising quality. Furthermore, tele-exercise, for instance, can be delivered through synchronous sessions, where patients interact with physical therapists in real time for guidance and feedback, or asynchronous formats, where pre-recorded instructional videos and digital exercise prescriptions are shared for patients to follow at their convenience. Asynchronous delivery is particularly cost-effective, reducing the burden on rehabilitation personnel while maintaining adherence through periodic progress tracking and check-ins. Both approaches can be tailored to the needs of older adults, ensuring accessibility without compromising the quality of care [137]. Telemedicine represents a practical and forward-thinking strategy to overcome the challenges inherent in managing the complex interplay of sarcopenia and MSK pain. By integrating digital tools into multidisciplinary workflows, healthcare providers can deliver holistic, accessible, and effective care to older adults, addressing the cyclical nature of these conditions.

## 10. Call to Action: Advancing Multidisciplinary Strategies

The complex interplay of sarcopenia and MSK pain underscores an urgent need for innovative, integrated care models to address their shared biological, psychological, and social mechanisms. While this perspective outlines actionable solutions, their implementation requires commitment across disciplines.

**To Clinicians:** Multidisciplinary collaboration must become a cornerstone of practice. Physical therapists, primary care physicians, dietitians, and mental health professionals should work together to ensure comprehensive screening, early diagnosis, and personalized interventions. As key healthcare team members, nurses play an essential role in implementing care plans, supporting adherence to nutrition and physical activity recommendations, identifying emerging psychosocial concerns, and promoting awareness and early referral for diagnostic evaluation. Practical steps include integrating sarcopenia assessments into routine MSK evaluations, fostering early identification through all points of care, and leveraging telemedicine to ensure continuity of care.

**To Policymakers and Health Systems:** Investments in education, infrastructure, and resources are critical. Priorities should include the following:Promote Multidisciplinary Collaboration: Develop frameworks and incentives to facilitate interdisciplinary teamwork among healthcare providers, ensuring holistic and integrated care for sarcopenia and MSK pain. This includes virtual care networks and shared treatment plans;Support Hybrid Care Models: Invest in telemedicine platforms and infrastructure to combine in-person care with tele-exercise and remote consultations, increasing accessibility and adherence while addressing workforce shortages;Invest in Professional Training: Provide funding for cross-disciplinary education programs to ensure healthcare providers understand the interconnected nature of sarcopenia and MSK pain, fostering a shared language and coordinated care strategies;Streamline Care Pathways: Introduce policies to integrate services across specialties, reducing fragmented care. This includes unified referral systems and electronic health records to enhance communication and continuity of care.

**To Researchers:** Future research must address critical gaps in understanding the interplay between sarcopenia and MSK pain, especially in community-dwelling older adults. Priorities include assessing the relationship between single or multisite MSK pain and sarcopenia-related parameters, identifying barriers to early diagnosis, and evaluating scalable interventions to bridge the gap between prevalence and treatment. Developing cost-effective diagnostic tools, refining multidisciplinary care models, and exploring the utility of telemedicine will be essential in mitigating the societal and economic burden of these conditions. Feasibility and implementation studies will also be critical to validating these real-world approaches.

## 11. Conclusions

The intersection of sarcopenia and MSK pain represents a pressing yet underappreciated challenge in aging populations. While these conditions are often approached in isolation, this perspective advocates for a paradigm shift, one that reframes these conditions as interdependent, requiring a unified, multidisciplinary response.

Breaking the cycle of mutual exacerbation between sarcopenia and MSK pain demands a transformation in how we deliver care. Beyond addressing physical symptoms, future approaches must embrace these conditions’ interconnected biological, psychological, and social dimensions. By leveraging hybrid care models, advancing telemedicine, and fostering collaboration across specialties, healthcare systems can bridge existing gaps and implement comprehensive, patient-centered solutions.

As populations continue to age, the opportunity to reshape care for sarcopenia and MSK pain is both timely and critical. We believe that prioritizing these strategies will improve individual health outcomes and contribute to the sustainability of healthcare systems, better equipping them to meet the complex needs of an aging society.

## Figures and Tables

**Figure 1 ijerph-22-00547-f001:**
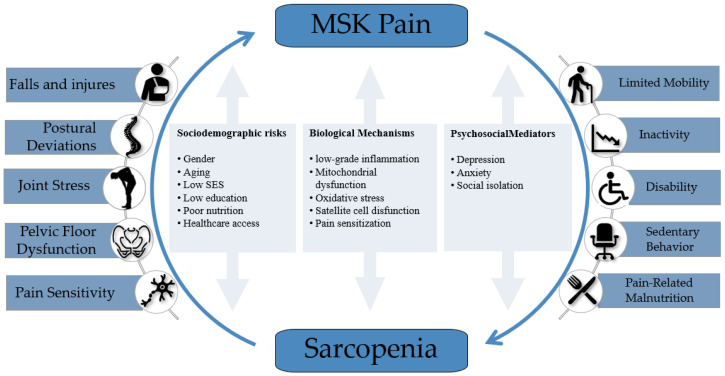
Bidirectional relationship between musculoskeletal (MSK) pain and sarcopenia. The diagram illustrates the cyclical consequences arising from each condition, along with shared sociodemographic risk factors, biological mechanisms, and psychosocial mediators that perpetuate the interplay. Although not depicted within the diagram, systemic health complications—such as cardiometabolic disorders, insulin resistance, and increased fat mass—may also act as downstream consequences and feedback factors that indirectly exacerbate both conditions.

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
