# Peer review of "The Intersection of Sarcopenia and Musculoskeletal Pain: Addressing Interconnected Challenges in Aging Care"

_ijerph, 2025, doi:10.3390/ijerph22040547_

Round 1
Reviewer 1 Report
Comments and Suggestions for Authors
The Intersection of Sarcopenia and Musculoskeletal Pain: Addressing Interconnected Challenges in Aging Care
The topic is relevant. However, I think it should be clear what type of literature review was carried out, the databases used and the criteria for including the articles. The abstract does not mention the main findings.
Is this research approach appropriate for this journal? If it's a review, can other reviews be cited?
Author Response
Comment: The topic is relevant. However, I think it should be clear what type of literature review was carried out, the databases used, and the criteria for including the articles. The abstract does not mention the main findings. Is this research approach appropriate for this journal? If it's a review, can other reviews be cited?"
Response: Thank you for your comments. We agree and therefore we have made the following changes:
- Clarification of Literature Review Type and Methodology:
Thank you for this comment. While this article was originally conceived as a review, it was subsequently adapted in accordance with the journal’s invitation and guidelines for a Perspective article. As such, the formal Methods section was removed to reflect the shift in article type. In response to your comment, we have now added a dedicated section titled “Approach and Scope of the Perspective” immediately following the Introduction. This section clearly describes the narrative approach taken, including the databases searched, the time frame, the search strategy, and the selection criteria used for article inclusion. We hope this clarification addresses your concerns and aligns with the journal’s expectations for transparency in literature-based syntheses. Page 2 Lines 65-91 - Abstract Adjustment (Main Findings):
As suggested, we revised the abstract to explicitly highlight the main findings of our perspective article. Page 1 lines 17-20 - Suitability of Approach and Citation of Other Reviews:
Thank you for raising this question. Our manuscript was specifically invited by the IJERPH Section Editor-in-Chief as a "Perspective" article. According to the journal's invitation, Perspectives highlight authors' views on emerging trends and future directions within a field, situated within recent literature. Thus, citing recent systematic reviews and meta-analyses is appropriate and necessary, as it situates our Perspective within the existing scientific context and facilitates meaningful discussion.
Reviewer 2 Report
Comments and Suggestions for Authors
This perspective article deals with the intersection of sarcopenia and musculoskeletal pain in an increasingly elderly population, providing a substantial worldwide problem in the future. The authors provide suggestions for future patient care and address gaps in research.
This is an interesting and well-written article. Thank you for that contribution.
I have some suggestions for improvement.
Pages 1 to 8 need to be reduced. It is a long read, and repetitions are losing the reader.
Especially section 2, where you first get to the point at the end.
Line 116 pp 3 I think you also need to address the use of the chair stand test, which is widely used across the world https://www1.racgp.org.au/ajgp/2024/october/sarcopenia-in-general-practice
In section 7, p 5, I think you need to address that chronic pain is easier for older people to get. I don´t think it is clearly stated.
I think it could be good for the readers if you visualized the cyclical relationship in a Figure.
I think the Implications for patient care could be elaborated on more. This section is the most interesting part, the one you are waiting for when reading the background.
I think you are missing a very important key care professional: the nurses. They often serve as the first point of contact with the elderly in their homes and at the hospitals and support them in exercising, eating protein-rich food, addressing depression and mental problems, etc.
In the paragraph “address psychosocial factors” page 10 you need to add that there is a well-known clear link between frailty/sarcopenia and cognitive impairment.
I think you need to perform extensive feasibility studies as a researcher to improve this situation.
Author Response
Comment: Pages 1 to 8 need to be reduced. It is a long read, and repetitions are losing the reader. Especially section 2, where you first get to the point at the end.
Response: Thank you for this comment. As this is an invited Perspective, the editorial invitation explicitly encouraged a structure similar to a review to highlight emerging trends with adequate context. Sections 1–5 were included to provide essential background for a multidisciplinary audience. While some aspects of the early content may be familiar to specialists, we believe this foundational information is important to support the central thesis of the Perspective. Nonetheless, we re-reviewed these sections and made targeted refinements to Section 3 to improve flow and remove redundancies. We hope this clarification and the revision in Section 3 address the reviewer’s concern while preserving the structure and intent of the invited Perspective. Page 2-3 lines 93-99
Comment: Line 116 pp 3 I think you also need to address the use of the chair stand test, which is widely used across the world https://www1.racgp.org.au/ajgp/2024/october/sarcopenia-in-general-practice
Response: Thank you for this observation. While we recognize the value of the chair stand test, particularly its endorsement by EWGSOP2, we intentionally did not include it as a commonly used strength measure across major diagnostic frameworks. Our paragraph aimed to summarize criteria shared by EWGSOP2, FNIH, AWGS, SDOC, and IWGS, where handgrip strength remains the primary and most consistently applied measure of muscle strength. The chair stand test is specific to EWGSOP2 and is not broadly incorporated across the other frameworks. For this reason, we chose to emphasize grip strength in this context. We hope this clarification addresses the reviewer’s concern (Voulgaridou et al., Nutrients 2024, 16, 436; https://www.mdpi.com/2662430).
Comment: In section 7, p 5, I think you need to address that chronic pain is easier for older people to get. I don´t think it is clearly stated.
Response: We thank the reviewer for this comment. In response, we have revised that section to explicitly state that older adults are more susceptible to developing chronic pain due to biological and physiological changes associated with aging. Page 5 lines 238-241
Comment: I think it could be good for the readers if you visualized the cyclical relationship in a Figure.
Response: We agree with this comment. Therefore, we have developed and included a new figure that visually illustrates the complex bidirectional and cyclical relationship between sarcopenia and MSK pain. Page 10
Comment: I think the Implications for patient care could be elaborated on more. This section is the most interesting part, the one you are waiting for when reading the background.
Response: Thank you for this comment. We agree that the Implications for Patient Care section is central to the article. To reflect this, the section was developed with three dedicated subsections—on screening, cyclical contributors, and telemedicine each offering concrete, multidisciplinary, and actionable strategies. We also included a separate Call to Action directed at clinicians, policymakers, and researchers. In response to additional reviewer comments, we further expanded this section to include the roles of nurses and cognitive impairment. We hope this comprehensive and structured approach addresses your expectation while maintaining the focus and clarity appropriate for a Perspective article.
Comment: I think you are missing a very important key care professional: the nurses. They often serve as the first point of contact with the elderly in their homes and at the hospitals and support them in exercising, eating protein-rich food, addressing depression and mental problems, etc.
Response: Thank you for this comment. While our original emphasis was on professionals directly involved in the diagnosis and structured management of sarcopenia, we fully agree that nurses play an essential role in supporting older adults across healthcare settings. In response, we have revised the “Call for Action” section to explicitly acknowledge their contribution to implementing care plans, supporting adherence, identifying psychosocial concerns, and promoting awareness and early referral for evaluation. We appreciate the opportunity to include this important perspective. Page 13 lines 607-612
Comment: In the paragraph “address psychosocial factors” page 10 you need to add that there is a well-known clear link between frailty/sarcopenia and cognitive impairment.
Response: Thank you for this comment. We have revised the paragraph under “Address Psychosocial Factors” to include the well-established association between sarcopenia, chronic MSK pain, and cognitive impairment, supported by recent literature. Pages 11-2 lines 533-539
Comment: I think you need to perform extensive feasibility studies as a researcher to improve this situation.
Response: Thank you for this comment. We agree that feasibility and implementation studies are essential to advancing and validating the proposed care strategies. We have added a sentence to the Call to Action for researchers highlighting the importance of evaluating real-world applicability alongside clinical effectiveness. Page 13 lines 637-638
Reviewer 3 Report
Comments and Suggestions for Authors
Dear Authors
The manuscript entitled, “The Intersection of Sarcopenia and Musculoskeletal Pain: Addressing Interconnected Challenges in Aging Care”, seems to me to be quite relevant from the point of view of the theme. The organization of the topics is pertinent, covering the most relevant aspects and is well-founded. The writing is objective and clear from the reader's point of view.
I believe that the intention was to carry out a (narrative) review of the literature, and in this sense my only constraint is the structure used.
Between topic 1 (introduction) and topic 2 (Skeletal Muscle: Significance and Age-Related Changes), in my opinion there is a missing cut-off point. Perhaps the placement of methodology (e.g.: Identification of the databases used in the research, the research period, keywords, etc.), or a title to make the separation (e.g.: development) and restart with a point one in this topic (1. Skeletal Muscle: Significance and Age-Related Changes).
Author Response
Comment: I believe that the intention was to carry out a (narrative) review of the literature, and in this sense my only constraint is the structure used.
Between topic 1 (introduction) and topic 2 (Skeletal Muscle: Significance and Age-Related Changes), in my opinion there is a missing cut-off point. Perhaps the placement of methodology (e.g.: Identification of the databases used in the research, the research period, keywords, etc.), or a title to make the separation (e.g.: development) and restart with a point one in this topic (1. Skeletal Muscle: Significance and Age-Related Changes).
Response: Thank you for your comment. As the article was originally developed as a review and later adapted to a Perspective format in response to the editorial invitation, we initially removed the formal Methods section. However, in response to this comment, we have now included a dedicated section titled “Approach and Scope of the Perspective” directly following the Introduction. This addition outlines the literature identification process, databases used, keywords, and inclusion criteria, and serves as a clear structural bridge between the introduction and thematic development of the article. Page 2 Lines 65-91
Round 2
Reviewer 1 Report
Comments and Suggestions for Authors
I consider that the authors have made important changes that have improved the quality of the manuscript